# Contribution to Characterizing the Meat Quality of Protected Designation of Origin Serrana and Preta de Montesinho Kids Using the Near-Infrared Reflectance Methodology

**DOI:** 10.3390/foods13101581

**Published:** 2024-05-19

**Authors:** Lia Vasconcelos, Luís G. Dias, Ana Leite, Etelvina Pereira, Severiano Silva, Iasmin Ferreira, Javier Mateo, Sandra Rodrigues, Alfredo Teixeira

**Affiliations:** 1Centro de Investigação de Montanha (CIMO), Instituto Politécnico de Bragança, Campus de Santa Apolónia, 5300-253 Bragança, Portugal; lia.vasconcelos@ipb.pt (L.V.); ldias@ipb.pt (L.G.D.); anaisabel.leite@ipb.pt (A.L.); etelvina@ipb.pt (E.P.); idasif00@estudiantes.unileon.es (I.F.); srodrigues@ipb.pt (S.R.); 2Laboratório Associado para a Sustentabilidade e Tecnologia em Regiões de Montanha (SusTEC), Instituto Politécnico de Bragança, Campus de Santa Apolónia, 5300-253 Bragança, Portugal; 3Department of Food Hygiene and Technology, University of Veterinary Medicine, Campus Vegazana S/N, 24007 León, Spain; jmato@unileon.es; 4School of Agriculture, Polytechnic Institute of Bragança, Campus de Santa Apolónia, 5300-253 Bragança, Portugal; 5Veterinary and Animal Research Centre (CECAV), Associate Laboratory of Animal and Veterinary Science (AL4AnimalS), University of Trás-os-Montes e Alto Douro, Quinta de Prados, 5000-801 Vila Real, Portugal; ssilva@utad.pt

**Keywords:** Serrana breed, Preta de Montesinho breed, kids’ goat, SVMR linear, prediction models, meat quality

## Abstract

The aims of this study were to describe and compare the meat quality characteristics of male and female kids from the “Serrana” and “Preta de Montesinho” breeds certified as “Cabrito Transmontano” and reinforce the performance of near-infrared reflectance (NIR) spectra in predicting these quality characteristics and discriminating among breeds. Samples of *Longissimus thoracis* (*n* = 32; sixteen per breed; eight males and eight females) were used. Breed significantly affected meat quality characteristics, with only color and fatty acid (FA) (C12:0) being influenced by sex. The meat of the “Serrana” breed proved to be more tender than that of the “Preta de Montesinho”. However, the meat from the “Preta de Montesinho” breed showed higher intramuscular fat content and was lighter than that from the “Serrana” breed, which favors its quality of color and juiciness. The use of NIR with the linear support vector machine regression (SVMR) classification model demonstrated its capability to quantify meat quality characteristics such as pH, CIELab color, protein, moisture, ash, fat, texture, water-holding capacity, and lipid profile. Discriminant analysis was performed by dividing the sample spectra into calibration sets (75 percent) and prediction sets (25 percent) and applying the Kennard–Stone algorithm to the spectra. This resulted in 100% correct classifications with the training data and 96.7% accuracy with the test data. The test data showed acceptable estimation models with R^2^ > 0.99.

## 1. Introduction

Goats have occupied an important place in human societies since the emergence of agriculture. They contribute significantly to both commercial and subsistence agriculture and are particularly prevalent in tropical, sub-tropical, and Mediterranean regions [1,2].

In Portugal, as in other Mediterranean countries, local or indigenous goat breeds such as the “Algarvia”, the “Bravia”, the “Charnequeira”, the “Preta de Montesinho”, the “Serpentina”, and the “Serrana” (the predominant) represent a cultural and valuable genetic heritage. It is essential to preserve and publicize them so that their value is maintained. However, local goat breeds may be less productive than introduced non-native breeds, so they need to be protected by certified quality labels, such as “protected designation of origin (PDO)”, “protected geographical indication (PGI)”, “traditional speciality guaranteed (TSG)”, “organic”, “free range”, or “grass-fed”, to foster their market position and address the demand for more reliable meats [3]. Products with these labels have to meet specific requirements for both composition and authenticity associated with meat quality traits. Moreover, in order to produce and guarantee certified quality meat, it is necessary to define the intrinsic quality characteristics of the meat and relate them to the production system, where origin and breed have a significant impact [3,4,5].

“Serrana” and “Preta de Montesinho” goats are major local breeds in Portugal. These goats are usually reared extensively, especially in mountainous areas of the regions of “Trás-os-Montes”, “Beira Interior”, “Ribatejo”, “Alentejo” and part of the “Algarve”, and “Minho”, with pasture, plateau, cork oak forest, and cereal stubbles, allowing them to make the best use of the natural environmental resources [6]. Four ecotypes of the “Serrana” goat have been classified at the national level: the “Jarmelista”, the “Ribatejano”, the “Serra”, and the “Transmontano” ecotypes, the latter being found in the northern interior of Portugal. These goats from those breeds are registered in the correspondent herd book according to the SPOC (Portuguese Sheep and Goat Shearing Society (SPOC)). Kid meat produced from goats of the “Serrana” and “Preta de Montesinho” breeds, according to the region, can obtain a PDO certificate, such as “Cabrito Transmontano”, or one out of five protected geographical indications (PGIs): “Cabrito Alentejano”, “Cabrito de Barroso”, “Cabrito da Beira”, “Cabrito da Gralheira”, and “Cabrito das Terras Altas do Minho” [7,8]. Protected animals can be purebred or crossbred, and the handling conditions or age at slaughter are set out in the specifications of each certification label. Their meat is highly appreciated for its distinct quality and soft texture [1,9,10]. To ensure that “Cabrito Transmontano” benefits from its PDO status and contributes to the preservation, cultural heritage, and genetic valorization of these local breeds, it is essential to study the characteristics responsible for the quality of the carcass and meat. Additionally, it is important to determine the extent to which these characteristics differentiate from more conventional goat breeds.

Spectrometric technologies are gaining importance as tools for carcass and meat quality control. They are non-destructive and non-invasive techniques that aim to provide data to assess the carcass composition and meat quality traits of animals. Practical aspects such as accuracy, reliability, cost, speed, and ease of use stand out. These technologies have recently attracted significant funding and research in an effort to improve food safety and quality standards in all areas of meat production.

Among those technologies, the ones that have deserved the most attention are near-infrared reflectance spectroscopy (NIRS), visible NIRS, hyperspectral imaging (HSI), computed tomography (CT), magnetic resonance imaging (MRI), and Raman spectroscopy (Raman) [11]. NIRS has made significant advances in its use, allowing its application in the assessment of meat quality attributes, with good prediction of meat composition [12,13,14,15]. There is limited recent research on the analysis of goat meat composition by NIRS compared to other meat species [13,14,16,17,18,19,20]. This fast, reliable, and non-destructive technology measures the absorption of electromagnetic radiation in the near-infrared spectrum (750–2500 nm) and requires a chemometric methods model able to relate the spectral absorption with analytical data obtained from a reference method of chemical properties [12,13]. NIR is combined with a multivariate statistical technique, such as principal component analysis (PCA), the projection algorithm (SPA), and the k-nearest neighbor algorithm (KNN), for quantitative and exploratory analysis. In addition, data models such as linear discriminant analysis (LDA), partial least square discriminant analysis (PLS-DA), and the support vector machine (SVM) can be applied for sample identification, characterization, and discrimination. The SVM, partial least square regression (PLSR), and the artificial neural network (ANN) are also used for quantitative purposes [11,17]. However, to develop reliable calibration models, a validation process is required for successful results. The validation method commonly used is internal or cross-validation or external validation, the latter of which can provide more reliable and relevant estimates of the model’s future predictive capacity; however, it is not advisable in biological samples or with a small number of samples [12]. Generally, the coefficient of determination (R^2^), root-mean-square error (RMSE), and/or ratio of performance deviation (RPD) are criteria generally used to predict meat quality [12].

The aims of this study were, on the one hand, to study goat meat from the breeds mentioned since its characterization will contribute to understanding and improving the quality of “Cabrito Transmontano”. On the other hand, the performance of the use of NIR-based methodologies to estimate the quality attributes of this important meat with protected designation of origin and to differentiate the meat between breeds has been evaluated.

## 2. Materials and Methods

### 2.1. Animal and Carcass Sampling

Kids from two local breeds from the “Trás-os-Montes” and “Alto Douro” regions were used in this study, as well as the “Serrana” and the “Preta de Montesinho” from herds, previously selected by a representative of the National Breed Producers Association of Serrana Goat (ANCRAS). The ANCRAS is a private organization that is responsible for the control, management, and certification of “Cabrito Serrano”, the Cooperative of Goat Producers of the Serrana Breed (CAPRISERRA, Crl) is the managing body of the respective “Protected Designation of Origin” (PDO), and SATIVA, Desenvolvimento Rural, Lda, is the private inspection and certification body. Animals were reared in a traditional extensive system with their mothers without any access to concentrates, hay, forage, or other supplements. When they reached the target age/slaughter weight established in the quality label regulation (slaughtered between 7 and 12 kg in weight and 60–80 days old) based on growth monitoring (daily gain; DG), they were separated from their mothers.

Kids were weighed immediately prior to slaughter (live weight at slaughter; LWS) at the Matadouro do Cachão, in Mirandela, a PDO-certified commercial slaughterhouse where they were slaughtered following standard commercial procedures according to the European normative for protection of animals at the time of killing [21]. Then, warm carcasses were weighed (hot carcass weight; HCW) and the weights were registered. After chilling for 24 h in a chamber at 4 °C, the carcasses were weighed again (cold carcass weight; CCW). The yield was calculated according to Equation (1) as follows:(1)$$yield=\frac{ccw}{lws}\times 100$$

The procedures were conducted on every sample individually. The research team purchased the carcasses of 32 kids from the slaughterhouse, 8 females and 8 males, making a total of 16 suckling kids per breed, and transported them to the Carcass and Meat Technology and Quality Laboratory of the Agrarian School of Bragança (LTQCC) for analysis. Carcass weights were provided by the slaughterhouse.

### 2.2. Physicochemical Analysis and Chemical Composition of Meat and Fat

The carcasses were split down the dorsal midline, and the *Longissimus thoracis* (LT) muscle portions were obtained between the 7th and 12th rib from each carcass. The characterizations were conducted on every sample individually.

#### 2.2.1. Value of pH and Instrumental Color

The pH value was measured on the left side of the *Longissimus lumborum* (LL) muscle between the 1st and 2nd lumbar vertebrae using a pH meter equipped with a Crison 507 penetrating electrode puncture (Crison Instruments S.A., Barcelona, Spain) following the standard method NP-ISO 3441/2008 [22]. Then, the samples were packed and refrigerated until further chemical composition analyses.

Instrumental color was evaluated on the cut surface of the right-side LT using a Lovibond RT Series—SP62 spectrophotometer (The Tintometer Limited, Wiltshire, England). The data were measured with a D65 illuminant at a 10° visual angle at 10 nm intervals with reflectance in the range of 400–700 nm in the CIELab coordinates [23]. The color of each LT was evaluated on the caudal surface (measured 45 min after cutting) for lightness (L*), redness (a*), and yellowness (b*). The chroma (saturation index—C*) and hue angle (H*) were also calculated according to Equations (2) and (3), respectively.
(2)$$C^{*}=\sqrt{\left({a^{*}}^{2}+{b^{*}}^{2}\right)}$$
(3)$$H^{*}={tan}^{-1}(\frac{a^{*}}{b^{*}})$$

A part of each LT of both sides was ground using a power mill Buchi Mixer B-400 (BÜCHI, Labortechnik AG, Postfach, Flawil, Switzerland) for around 5 to 10 s to obtain a homogeneous paste (weighing around 100 g). The rest of the LT muscle samples were used for the water-holding capacity (WHC) and shear force (SF) analysis.

#### 2.2.2. Moisture, Protein, and Ash Content

The determination of moisture was performed according to NP-ISO-1614/2002 [24]; approximately 3 g of the sample was weighed, and 5 mL of ethanol (96% *v*/*v*) was added to the sample. After that, samples were oven-dried (Raypa DO150, Barcelona, Spain) for 24 h at 103 ± 2 °C, and the weight lost was determined.

The protein content was analyzed using the Kjeldahl method (Kjeldahl Sampler System (K370 and Digest System K-437; Flawil, Switzerland) according to NP-ISO-1612/2002 [25]. All values were expressed in percentage (g/100 g of meat). For ash content, according to NP-ISO-1615/2002 [26], the samples were incinerated at 550 °C ± 25 °C during 5–6 h in a muffle furnace (Vulcan BOX Furnace Model 3-550, Yucaipa, CA, USA), and the mass of ash obtained was measured.

#### 2.2.3. Water-Holding Capacity and Shear Force

For the WHC and SF, according to the Honikel procedure [27], samples of the LT muscle (approximately 100 g) were cooked inside plastic bags in an 80 °C water bath until reaching 70 °C core temperature and were measured with a Therma 3 K-type penetration temperature probe (Electronic Temperature Instruments Ltd., Worthing, United Kingdom). The samples were cooled at room temperature for 30 min and weighed. Then, the steaks were kept overnight in a chamber at 4 °C, taken out of the cold, and left to stand until they reached room temperature for 60 min. A total of 5 muscle sub-samples (1 cm^2^ cross-section) were taken from each LT muscle portion for SF evaluation, which was determined using an INSTRON 5543J-3177 (Instron Limited, Cerdanyola, Spain) equipped with a Warner–Bratzler device. The measurement was recorded as the average yield force in kilograms (kgf) required to shear perpendicularly to the fiber direction at a cross-head speed of 3.33 and 1 mm s^−1^ [23].

#### 2.2.4. Total Intramuscular Fat and Fatty Acid Profile

The total intramuscular fat (IMF) content of the LT sample was extracted from 25 g according to the Folch procedure and weighted [28]. The fatty acid (FA) profile was determined using approximately 50 mg of the fat. Once obtained, FAs were transesterified according to Domínguez et al. [29]. The FA methyl ester separation and quantification were performed using a gas chromatograph (GC-Shimadzu 2010Plus; Shimadzu corporation, Kyoto, Japan) and a Supelco SPTM-2560 (Supelco, Bellefonte, PA, USA) fused silica capillary column (100 m × 0.25 mm × 0.2 μm film thickness). The content of FAs was expressed as the g of FA/100 g of total FAs. The nutritional quality of FAs was determined by the n-6/n-3 and polyunsaturated/saturated FA (PUFA/SFA) ratios [30], and the thrombogenicity (IT) and atherogenicity (IA) indexes were calculated as indicated in Equations (4) and (5) and described by Ulbricht and Southgate [31]. The hypocholesterolemic and hypercholesterolemic fatty acid (h/H) ratio was also calculated, as indicated in Equation (6), according to Santos-Silva et al. [32].
(4)IA=C12:0+4×C14:0+C16:0∑MUFA+∑PUFA(5)IT=C14:0+C16:0+C18:00.5×∑MUFA+0.5×∑PUFA n-6+3×∑PUFA n-3+PUFA n-3PUFA n-6(6)hH=C18:1n-9+C18:2n-6+C20:4n-6+C18:3n-3+C20:5-n3+C22:5n-3+C22:6n-3C14:0+C16:0

All analyses were performed in triplicate for each method and each animal.

### 2.3. Near-Infrared Spectroscopy

A total of 32 samples of LT were minced individually using a power mill Buchi Mixer B-400 (BÜCHI, Labortechnik AG, Postfach, Flawil, Switzerland), and each one was placed in Petri dishes with a diameter of approximately 9 cm. An FT-NIR Master^TM^ N500 (BÜCHI, Labortechnik AG, Postfach, Flawil, Switzerland) was used for spectral analysis. The instrument operates within a spectral range of 4000 to 10,000 cm^−1^ (1000 to 2500 nm), with a resolution of 4 cm^−1^ and a 360° rotation system. Three spectra were measured for each 32 LT sample, and the average of these spectra was used for further chemometric analysis. These repetitions were aimed at accommodating the intrinsic variability within meat samples. This is especially critical considering the technique’s aim of enabling in situ meat analysis on processing lines, where variability must be incorporated into the modeling process.

### 2.4. Statistical Analysis

Data obtained for all meat quality traits were subjected to a one-way analysis of variance (ANOVA) using the generalized linear model (GLM) procedure with the factors of sex, breed, and their interaction, and the level of significance was set at * 0.05, ** 0.01, or *** 0.001.

### 2.5. Chemometrics and Data Analysis

Version 5.5 of NIRCal BÜCHI software was applied to save all spectra in an Excel^TM^ file. For data treatment, R (application Intel, R.app GUI 1.78, © R Foundation for Statistical Computing, 2021) and RStudio (application Intel, RStudio 2022.02.3 Build 492, © 2009–2022 RStudio, PBC) were used. The independent data correspond to the NIRS spectrum obtained from the LT samples analyzed. The three spectra replicates were used in the analysis since it allowed us to include the variability associated with heterogeneity of the samples contributing to the adjustment of more robust models.

To remove infrared scattering effects and increase their variations [33], the smoothed spectra were pre-treated with six different methods: normalization of each line to the unit area (NORM), baseline correction with asymmetric least squares (ALS), multiplicative scatter correction (MSC), standard variable normalization (SNV), first derivative (DV1), and second derivative (DV2). Combinations of NORM, ALS, MSC, and SNV with DV1 were also considered [34].

Each spectrum represents a set of correlated data whose number exceeds the number of samples analyzed, and a selection of variables was made to reduce their information to 10%. To perform this, points with a wavenumber range of 20 cm^−1^ were selected, allowing the spectrum to be reduced from 1501 points to 151 points.

The sample spectra were divided into calibration sets (training group—twenty samples (75% of samples), and each was subjected to three repetitions of NIR spectral analysis conducted from three distinct positions in order to encompass the variability inherent in meat sample), and prediction sets (test group—10 samples (25% of samples) with three spectrum repetitions of NIR analysis). Data partitioning was conducted using the Kennard–Stone algorithm with the Euclidean metric, leveraging the physicochemical attributes of the meat samples [35,36] to delineate these two distinct data groups, common across all data treatments. These two groups were used in two data treatments (discrimination and quantification) by applying the linear support vector machine (SVM) techniques that do not depend on the distributions of the underlying dependent and independent variables.

The first treatment involves the discrimination of meat samples from the two breeds (dependent variable) under study based on the linear SVM classification model using smoothed NIR spectra and the first derivative (independent variables). The classification model was selected based on the highest value of correct classifications (accuracy) in both the training and testing groups.

The support vector machine (SVM) technique does not depend on the distribution of the underlying dependent and independent variables.

The second treatment evaluates NIRS capability to quantify each meat quality parameter (dependent variables) with a linear SVM regression model selected within the several considered treatments applied to the spectra. The best model was selected considering the lowest root-mean-square error (RMSE), and, as a second criterion, the highest value of determination coefficient and slope obtained from the relation established between the predicted model and real values for train and test data groups. The results were considered acceptable if the linear regression parameters were close to the theoretical values [37,38]: “zero” (0) for relative standard error (RSE) and intercept, and “one” (1) for slope and the determination coefficient.

## 3. Results and Discussion

### 3.1. Animal and Carcass Sampling

Table 1 shows the mean values for some of the carcass characteristics of the kids from the breeds studied, namely, age at slaughter, DG, LWS, CCW, and yield. No significant differences (*p* > 0.05) were found between breeds and between sexes, except for the LWS (*p* ≤ 0.05), because the weight of the “Preta de Montesinho” kids was slightly lower. This difference may be due to variations in the commercial practices of the farms from which the animals originated, although it was still within the above-mentioned age and weight ranges stipulated in the specifications for “Cabritos Transmontanos” by ANCRAS.

The daily weight gain (DG) ranged from 110 to 119 g/day. Slightly higher values were registered for the kid of six goat breeds, “Boer Angora”, “Boer Feral”, “Boer Saanen”, “Feral Feral”, “Saanen Angora”, and “Saanen Feral”, by Dhanda et al. [39], with values between ≈ 126 g/day and ≈167 g/day. The carcass yield obtained was in the range found for kids of similar ages. However, our values were slightly higher than those observed by Teixeira et al. [9] (a mean of 50% and a range between 46.49% and 54.16%) and in “Cabritos Transmontano” animals of the same weight as those used in this study. Contrary to our findings, other authors have found differences in carcass yield of kids associated with breed [40] or sex [41]. The latter studied animals of the “Bravia” breed at 12 weeks of age and found yields of 55.9% for males and 51.7% for females.

### 3.2. Physicochemical and Chemical Composition of Meat

Table 2 shows the mean values and the standard error of the means for the meat quality traits according to breeds and sexes. It could be observed that breed was the predominant factor affecting the quality traits, and sex only influenced color.

The pH of the LT muscle showed that there was no influence of breed or sex on this parameter. In agreement, Teixeira et al. [9] and Santos et al. [42] also found no differences between sex in “Serrana” and “Barroso” breed goat kids, respectively, in terms of pH measured over 24 h. In line with our results are the values obtained by other authors, such as Argüello et al. [43] (≈5.68 for ”Majorera” with 10 kg LWS without age), Santos et al. [10] (≈5.88 for “Serrana” and ≈5.67 for “Bravia” with 8–11 kg of LWS both), and Ripoll et al. [44] (≈5.86 for “Majorera”, ≈5.85 for “Palmera”, and ≈5.88 for “Tinerfeña”). The pH obtained in our study can be considered high but within an acceptable range (from 5.5 to 6.2) [14,45,46]. Authors such Dhanda et al. [39] argue that pH values in goat meat close to or above 6.0 are indicators of animal stress before slaughter. Generally, kids of goats are more sensitive and susceptible to stress-depleting muscle glycogen than other species, and low glycogen at slaughter is a common (not only) cause of high pH in goat meat [47]. Another reason for the high pH is that the very small carcasses of kids are cooled very quickly in the slaughterhouse cold stores, which makes post-mortem glycolysis very slow and incomplete [48]. Although, in some regions of the world, it is preferred to consume meat from young animals [9,49]; early age at slaughter can cause severe stress, which results in a higher pH and, consequently, negatively affects texture, color defects, etc. [50].

Goat meat was reported to have a superior WHC compared to lamb [51]. The pH and the protein content of meat play a fundamental role in the levels of expelled exudation (WHC). In this study, the “Serrana” breed showed significantly lower cooking losses (higher WHC) than the “Preta de Montesinho” breed (*p* < 0.001). The difference between breeds could be related to differences in composition, i.e., lower intramuscular fat content. Sex did not affect the WHC (*p* > 0.05). In contrast, Todaro et al. [48] found for the “Nebrodi” breed that cooking losses tend to be higher in meat from female kids.

There were significant differences in shear force (Warner–Bratzler) between breeds (higher in “Preta de Montesinho” than in “Serrana”; *p* < 0.001) but not between sexes. This difference can be explained by the higher WHC of “Serrana” meat and, therefore, the lower myofibrillar density in cooked “Serrana” meat. In meat from young animals, the effect of collagen on toughness is small. SF values between 2 and 3 kgf were found by Ripoll et al. [44] in local Spanish “Majorera”, “Palmera”, and “Tinerfeña” breeds. However, higher values of ≈ 5.8 kgf were found for 10 kg LWS “Majorera” breed kids fed milk by Argüello et al. [43] and Coelho [41] in “Bravia” breed kids, without differences in sexes (≈5.6 kgf for males and ≈6.0 kgf for females). Webb et al. [49] argued that shear forces higher than ≈4.9 kgf might not be acceptable to consumers, so our values are within the acceptable range. It was noticed that with the increased body weights, SF values dropped notably (higher carcass weights in “Serrana” (Table 1) and were lower compared to the “Preta de Montesinho” breed (Table 2)) with the matured carcass weight. Cutting forces were also reduced, contrary to other studies that confirmed higher SF values with higher carcass weights, especially in female goats [10,43]. Due to the greater deposition of body fat that this sex inherently implies, cutting forces were significantly lower, thus indicating more tenderness and flavorful meat. This was also proven by Teixeira et al. [52] and Ripoll et al. [44].

Muscle color is another important trait to assess carcass and meat quality that influences the consumer’s willingness to purchase. The impact of the breed was high; a* and b* (*p* ≤ 0.01) were affected, and thus was C* (*p* < 0.001). Redness and color intensity were higher in “Serrana” meat. Gawat et al. [5] and Peña et al. [41] observed that breed has an effect on redness values (a*) of kid meat. The influence of genotype on the color attributes of kid meat is still under debate because it can be confounded with factors related to feed and weight. The variations in a* in fresh meat are often correlated with the quantity of heme pigments presented in muscles [10,42,53]. The Fe content of meat in young kid animals varies with feeding [45]. Meat color in young animals increases after feeding. Results suggest that “Serrana” kids tended to eat less milk and more pasture during the growing period. This can also be related to the difference in LWS. Heavier body weight of kids at slaughter results in darker and more red meat [43].

With regard to moisture, protein, and ash parameters, there was no significant influence of breed or sex (*p* > 0.05). In terms of moisture content, the kid meat showed values of around 75%, which is to be expected given that such young animals show little fat in the carcass. Moisture content similar to or higher than 74% has been found in other studies, such as Ripoll et al. [44], with ≈74.07% in “Palmera” and ≈74.66% in “Tinerfeña” breeds; Coelho [41], with ≈75.89% values for the “Bravia” breed; and Argüello et al. [43], with percentages of around 78 for “Majorera” kids. High moisture contents in kid meat are associated with high protein content, i.e., around 20% [44].

Intramuscular fat (IMF) was affected by breed (*p* ≤ 0.05). The LT muscle from “Serrana” kids had a lower IMF percentage compared to “Preta de Montesinho” kids. In agreement with these results, Ripoll et al. [44] reported the level of influence on the breed to be ≈1.29% for “Palmera” and ≈1.97% for “Florida”; and for the authors Peña et al. [41] in their study, 1.18% for “Criollo Cordobes” and 1.32% for “Anglonubian” intramuscular fat Also, Quaresma et al. [8] verified that certified local Portuguese breeds “Cabrito”, “Minho”, “Barroso”, and “Transmontano” displayed lower fat contents than “Sannen”, a foreign breed (0.86 versus 1.91 g/100 g of meat). Argüello et al. [43] presented an IMF value of 1.33 ± 0.57%, and Rodrigues and Teixeira [2] presented a value of 2.12%, which was lower than those obtained from weaned “Pateri” goats in a study by Talpur et al. [54]. It can thus be observed that the breeds, due to precocity and feeding practices of the animals, are two factors that affect IMF content. However, a low amount of IMF, near 1% and lower than 2%, is characteristic in suckling goat kids [43,44,55] (around 0.84–1.26%), which is characteristic of young unweaned animals. In this study, sex did not affect IMF (*p* > 0.05), although some authors, such as Todaro et al. [48] and Santos et al. [10], argue that female kids tend to have increased fat deposition compared to males.

### 3.3. Fatty Acid Profile and Lipid and Lipidic Quality

Table 3 shows the fatty acid profile and the total concentrations of saturated fatty acids (SFAs), monounsaturated fatty acids (MUFAs), and polyunsaturated fatty acids (PUFAs) present in the LT muscle of “Serrana” and “Preta de Montesinho” kids.

In the meat studied, the percentages of SFA and MUFA were similar to each other, representing 90% of the total FA, with 10% PUFA. These values are comparable to those found by Ripoll in the meat of suckling kids reared on breast milk and agree with the typical FA composition of suckling kid meat [10,56]. No significant differences by either breed or sex were found in any of the FA sums or ratios. The lack of differences can be attributed to a similar rearing system and similar intramuscular fat content for the two breeds.

The ratio of n-6/n-3 PUFA was close to 7:1. The n-6/n-3 ratio is an indicator of the healthiness of fats in relation to heart disease and diabetes. A low ratio of n-6/n-3 fatty acids in the diet is desirable in terms of prevention of those diseases, and it is considered that the appropriate ratio should be less than four [57,58] and preferably close to one [59,60]. This is comparable to what was reported by Ripoll et al. [56] and Horcada et al. [61] but higher than what was reported by Talpur et al. [54]. The atherogenic (IA) and thrombogenic (IT) indices are correlated with the amounts of SFA, MUFA, and PUFA without a ratio recommendation [62]. The values obtained in this study for both indices were similar to Teixeira et al. [63] and Wood et al. [64]. With respect to the h/H ratio, the results were also acceptable.

The predominant SFA found in the meat studied were, in order, palmitic acid (C16:0), stearic acid (C18:0), and myristic acid (C14:0). Individual SFAs were not affected by breed or sex, with the exception of C24:0, which was significantly affected by breed (*p* ≤ 0.01), and C12:0 was affected by sex (*p* ≤ 0.05). In contrast, Ripoll et al. [56] found breed to have a significant effect on SFA percentages in suckling kid meat. Regarding MUFA percentages, the most abundant were oleic acid (C18:1n-9) and palmitoleic acid (C16:1n-7); breed only affected the percentages of C20:1n-9 and C15:1, and no effect of sex was found. This differs from the findings of Santos et al. [10], who found that MUFA was influenced by sex. The most abundant PUFAs were linoleic acid (C18:2n-6), arachidonic acid (C18:4n-6), and linolenic acid (C18:3n-3). No PUFAs were affected by breed or sex.

### 3.4. Qualitative and Quantitative Predictive Model

To try to understand what information NIR may be using to make this separation, modeling was carried out between the spectra (independent variables) and the variables referred to in Table 2 and Table 3 (dependent variables). The information presented below refers to the models with predictive capacity in the test data, and it is important to note that acceptable estimation models were obtained (R^2^ > 0.99) for the other dependent variables, which may be indicative that with more samples there is the possibility of also obtaining forecast models. So, regarding the quantitative SVM model, the best model for the relation between dependent variables and the different treatment on spectra (independent variables; Appendix A) was selected by tunning the best C value in the model, which gave the lowest root-mean-square error (RMSE) values. A second selection criteria was the coefficient of determination (R^2^), selecting the best model that gave R^2^ _train_ > 0.95 and slope _train_ > 0.95 (Table 4). Within those possible selected models, the final selection considered the results with the test data: a lower RMSE and R^2^ _test_ > 0.90, and the results are presented in Table 5. The SVMR technique proved to be suitable for modeling meat characterization data (Table 4 and Table 5). The slope and intercept values presented in these tables are significant, with *p*-values lower than 0.025; but, in two cases in Table 4 and all cases in Table 5, the intercept was assigned as not significant (ns; *p*-value > 0.05).

For the pH model with the transformation of the spectra (NORM-DV1), the following training parameters were acceptable: R^2^ = 0.991 and RMSE = 0.016. A small ordinate at the origin had acceptable predictive performance in the test data (99.86% of the overall variability was explained). The protein model also showed acceptable training parameters of R^2^ = 0.986, RMSE = 0.116, an intercept of 0.935 ± 0.293, and test group data predictive performance (99.75% of test data variability explained). Although these parameters were not able to differentiate the breeds in the first analysis, they were differentiators in the prediction within the group of animals. Li et al. [18] obtained partial least square regression values of R^2^ = 0.89 and RMSE = 0.03 in the calibration train data and R^2^ = 0.79 and RMSE = 0.04 in prediction test data for the pH parameter in sheep meat. Also, Vasconcelos et al. [17] presented a support vector machine regression polynomial function for protein quantification in an indigenous pork breed with similar results: an R^2^ of 0.993 and 0.998 and an RMSE of 0.089 and 0.940 with regard to training and test sets, respectively.

On the contrary, the color parameters L*, a*, b*, and C* (with the exception of H* for NIR) stand out for allowing discrimination between the two breeds. It was possible to establish NIR predictive models for those color parameters, inferring that this information can be used by NIR to discriminate between the two breeds (Table 5). Values of R^2^ > 0.96 and slope > 0.96 for L*, a*, b*, and C* were obtained. Li et al. [18] also analyzed color parameters and obtained good model performance but low predictive ability, i.e., lower than that recorded in this work, for L*(R^2^ = 0.78; RMSE = 1.80), a* (R^2^ = 0.68; RMSE = 0.71), and b* (R^2^ = 0.75; RMSE = 0.71).

Models with predictive performance (R^2^ > 0.90, RMSE > 0.11 and slope > 0.83 in test data) were obtained for parameters C14:0, C16:0, C16:1n-7, C17:1n-7, C18:0, C18:0, 9t-C18:1, C18:1n-9, C18:3n-3, ∑MUFA, IA, and h/H. The accuracy of the prediction models varied with the FA content. The predictions were satisfactory for FA groups or individual FA present at medium to high concentrations (total MUFA, and C14.0, C16:0, C18:0, C18:1n-9) but were lower for FA generally found in meat at low or very low concentrations (C18:1 9 trans, C18:3n-3, C17:1n-7, C16:1n-7).

As the main constituents are more easily predictable by NIR spectroscopy than compounds with low concentrations, increasing the concentration range would result in fewer limitations.

Predicting the composition of the FA in the meat of different species, as well as classifying it based on geographical origin, has already been studied using NIR spectroscopy. Guy et al. [15] reported models for predicting lamb spectra (cross-validation R^2^ = 0.98 versus 0.53) for FA groups. They observed that the heterogeneous nature of the muscles probably limited the ability to generate more accurate calibration models for FA contents based on NIR spectra. Pullanagari et al. [65] observed low prediction accuracies for individuals and groups of FAs when NIR spectra were collected on intact longissimus lumborum from lambs. They not only attributed the cause to the heterogeneity of the sample but also the lack of success due to the low efficiency of fat extraction for reference samples. Also, Teixeira et al. [14] and Sun et al. [66] reported that NIR spectroscopy had great accuracy in goat and lamb meat, respectively, from samples from mountain regions since NIR spectroscopy combined with chemometrics was able to effectively characterize meat by geographical origin.

Finally, with the purpose of evaluating the performance of NIR spectra in discriminating meat samples from the two breeds, a discriminant analysis was conducted using the linear SVM technique applied to different treatments on spectra (independent variables; Appendix A). The selected estimation model was obtained (with the tuning of the best cost and coef0 model parameters set to one and zero, respectively) using spectra that combined SNV and first derivative transformations. This approach resulted in 100% correct classifications with the training data and 96.7% accuracy with the test data. Only one repetition of a sample was misclassified, demonstrating the robustness of NIR analysis despite the high variability inherent in dairy goat muscle. This variability is reflected in NIR analysis through the acquisition of spectra from three different positions. So, the average overall sensitivity and specificity in both groups was 100%. These results showed the effectiveness of NIR analysis in classifying the LTL muscle of dairy goats of the “Serrana” and “Preta de Montesinho” breeds. Considering the global results obtained, it is expected that the robustness of the discriminant model can be improved if there is a greater representability of natural samples, ensuring more variability in training data. In this way, the results obtained will be more accurate in animals where specific and rigorous quality assessment criteria result in products with quality seals.

## 4. Conclusions

This study has characterized the quality traits of suckling kid meat from two Portuguese breeds with a protected designation of origin from European quality labels, namely, “the Serrana” and “Preta de Montesinho”. Differences in key quality characteristics such as intramuscular fat, cooking losses, hardness, and redness have been observed between both breeds, despite the young age of the animals and the genetic and geographical closeness of the breeds. Meat from the “Serrana” breed showed the advantage of retaining more water and being, therefore, more tender than the “Preta de Montesinho”. However, the “Preta de Montesinho” breed has a lighter meat color than the “Serrana”, which favors its color quality. Regarding the significant differences in intramuscular fat content (which is higher in “Preta de Montesinho”), while this fat level is generally considered a quality factor in ruminant meat, its impact is less clear in the case of suckling kid meat, which typically has low levels of intramuscular fat. In contrast to breed, sex does not seem to have an important role in quality characteristics. This study has also highlighted the use of cutting-edge methodologies, such as NIR, to estimate the quality attributes of the meat of these animals. NIR can effectively play an important role in maintaining consistent product quality. It could be observed that breed was the predominant factor affecting the quality traits, while sex only influenced color. There were no differences between breeds or sexes regarding fatness. The overall results enabled the quantification and prediction of meat quality characteristics, which is promising for the individual characterization of animals in breeding programs (free-range system). This adds value to a product with the specificity of a protected designation of origin. In the future, by including a larger number of samples of these breeds, it will be possible to expand the range of spectra measured, thereby increasing the number of parameters to be quantified and enhancing the estimation and prediction capacity of the models. This study supports the distinctiveness and marketing of the “Serrana” and “Preta de Montesinho” breeds under a quality label, protecting them in a competitive market.

## Figures and Tables

**Table 1 foods-13-01581-t001:** Effect of breed and sex on traits of “Serrana” and “Preta de Montesinho” kid goats.

Traits	Serrana		Preta^1^		SEM	Significance		
	Males	Females	Males	Females		Breed	Sex	Breed × Sex
Age (days)	72.4	73.6	73.1	67.0	5.012	ns	ns	ns
DG (g/day)	118.3	119.4	110.6	110.9	11.51	ns	ns	ns
LWS (kg)	10.9 ^ab^	10.8 ^ab^	10.2	9.2	0.559	*	ns	ns
CCW (kg)	5.8	6.0	5.7	5.1	0.325	ns	ns	ns
Yield (%)	53.2	55.5	55.8	55.4	1.157	ns	ns	ns

ns—not significant (*p* > 0.05); * (*p* ≤ 0.05); SEM—standard error of the mean; ^ab^—significant differences within same row; Preta^1^—Preta de Montesinho; DG—daily gain; LWS—live weight at slaughter; CCW—cold carcass weight.

**Table 2 foods-13-01581-t002:** Effect of breed and sex on physicochemical meat characteristics of “Serrana” and “Preta de Montesinho” kid goats.

Parameters	Breed		SEM	Sex		SEM	Significance		
	Serrana	Preta^1^		Male	Female		Breed	Sex	Breed × Sex
Physical parameters									
pH (24 h)	5.98	5.88	0.041	5.98	5.88	0.04	ns	ns	ns
WHC (%)	12.93	19.23	1.222	15.23	16.91	1.255	***	ns	ns
SF (kgf)	2.33	3.88	0.156	3.27	2.94	0.161	***	ns	ns
Color parameters									
L*	54.71	52.56	0.869	55.17	52.09	0.892	ns	*	ns
a*	18.39	14.70	0.611	16.47	16.62	0.628	**	ns	ns
b*	12.96 ^ab^	11.42 ^ab^	0.257	12.39 ^ab^	11.99 ^ab^	0.264	**	ns	*
H*	35.43	38.03	0.860	37.40	36.07	0.884	*	ns	ns
C*	22.54	18.65	0.584	20.67	20.52	0.599	***	ns	ns
Chemical parameters									
Moisture (%)	75.80	74.78	0.530	75.58	75.00	0.545	ns	ns	ns
Ash (%)	1.20	1.22	0.086	1.21	1.20	0.088	ns	ns	ns
Protein (%)	21.06	21.66	0.254	21.37	21.34	0.261	ns	ns	ns
IMF (%)	1.20	1.81	0.162	1.47	1.54	0.167	*	ns	ns

ns—not significant (*p* > 0.05); * (*p* ≤ 0.05); ** (*p* ≤ 0.01); ***(*p* < 0.001); SEM—standard error of the mean; ^ab^—significant differences within same row; Preta^1^—Preta de Montesinho; WHC—water-holding capacity; SF—shear force; (L*)—luminosity index; (a*)—red index; (b*)—yellow index; (C*)—chroma; (H*)—tom; IMF—intramuscular fat content.

**Table 3 foods-13-01581-t003:** Fatty acid profile of intramuscular fat of the LT muscle from the “Serrana” and “Preta de Montesinho” kid goat breeds. Effect of breed and sex.

FA (%)	Breed		SEM	Sex		SEM	Significance		
	Serrana	Preta^1^		Male	Female		Breed	Sex	Breed × Sex
C10:0	0.114	0.090	0.014	0.113	0.092	0.014	ns	ns	ns
C12:0	0.483	0.424	0.038	0.511	0.396	0.039	ns	*	ns
C14:0	5.435	4.564	0.359	5.298	4.700	0.369	ns	ns	ns
C14:1	0.206	0.176	0.026	0.203	0.179	0.206	ns	ns	ns
C15:0	0.211	0.211	0.027	0.223	0.219	0.028	ns	ns	ns
C15:1	0.163	0.121	0.011	0.151	0.132	0.011	**	ns	ns
C16:0	26.198	25.032	0.573	25.681	25.549	0.588	ns	ns	ns
C16:1n-7	2.300	2.314	0.130	2.278	2.336	0.133	ns	ns	ns
C17:0	0.765	0.738	0.043	0.759	0.745	0.044	ns	ns	ns
C17:1n-7	0.592	0.585	0.032	0.575	0.601	0.033	ns	ns	ns
C18:0	11.682	11.935	0.487	12.253	11.365	0.500	ns	ns	ns
9t-C18:1	1.637	1.592	0.093	1.7001	1.530	0.096	ns	ns	ns
C18:1n-9	38.749	38.666	0.984	37.602	39.812	1.010	ns	ns	ns
9t, 12t-C18:2	0.281	0.275	0.020	0.252	0.304	0.021	ns	ns	ns
C18:2n-6	6.489	6.614	0.607	6.817	6.287	0.623	ns	ns	ns
C20:0	0.045	0.027	0.007	0.034	0.038	0.007	ns	ns	ns
C18:3n-6	0.066	0.069	0.010	0.070	0.065	0.010	ns	ns	ns
C20:1n-9	0.439	1.048	0.085	0.717	0.770	0.087	***	ns	ns
C18:3n-3	0.665	0.679	0.042	0.688	0.657	0.043	ns	ns	ns
C21:0	0.012	0.010	0.006	0.0076	0.014	0.006	ns	ns	ns
C20:2n-6	0.042	0.027	0.008	0.036	0.033	0.009	ns	ns	ns
C22:0	0.169	0.237	0.046	0.196	0.210	0.047	ns	ns	ns
C20:3n-6	0.141	0.198	0.029	0.164	0.175	0.029	ns	ns	ns
C22:1n-9	0.015	0.023	0.008	0.016	0.022	0.008	ns	ns	ns
C20:4n-6	2.339	2.387	0.335	2.520	2.205	0.344	ns	ns	ns
C20:3n-3	0.044	0.004	0.022	0.007	0.040	0.022	ns	ns	ns
C22:2n-6	0.000	0.002	0.000	0.002	0.001	0.000	ns	ns	ns
C24:0	0.214	0.865	0.141	0.451	0.628	0.145	**	ns	ns
C24:1n-9	0.231	0.402	0.116	0.268	0.365	0.119	ns	ns	ns
C22:6n-3	0.234	0.670	0.135	0.511	0.511	0.138	*	ns	ns
Σ SFA	45.355	44.139	0.852	45.532	43.961	0.875	ns	ns	ns
Σ MUFA	44.335	44.931	0.999	43.515	45.751	1.025	ns	ns	ns
Σ PUFA	10.308	10.929	1.059	10.952	10.286	1.087	ns	ns	ns
PUFA n-6	9.025	9.299	0.645	7.097	6.604	0.663	ns	ns	ns
PUFA n-3	1.222	1.354	0.436	3.602	3.377	0.448	ns	ns	ns
PUFA n-6/n-3	7.385	6.867	0.233	2.054	2.286	0.239	ns	ns	ns
IA	0.892	0.797	0.045	0.881	0.808	0.04	ns	ns	ns
IT	1.225	1.1528	0.057	1.215	1.162	0.059	ns	ns	ns
h/H	1.538	1.688	0.072	1.575	1.651	0.074	ns	ns	ns

ns—not significant (*p* > 0.05); * (*p* ≤ 0.05); ** (*p* ≤ 0.01); *** (*p* < 0.001); SEM—standard error of the mean; ^ab^—significant differences within same row; FA—fatty acid; Preta^1^—Preta de Montesinho; SFA—saturated fatty acid; MUFA—monounsaturated fatty acid; PUFA—polyunsaturated fatty acid; PUFA n-6/n-3 (∑ omega-6) (∑ omega-3); IA—index of atherogenecity; IT—index of thrombogenicity; h/H—hypocholesterolemic/hypercholesterolemic index. Only fatty acids that represented more than 0.1% are presented in the table, although all detected fatty acids were used for calculating the totals and the indices.

**Table 4 foods-13-01581-t004:** Parameters of the obtained linear support vector regression model and performance estimation on the training data.

Parameters	Data Treatment	Best C	RMSE	R^2^	Slope ± s *	Intercept ± s
Physical parameters						
pH (24 h)	NORM-DV1	10	0.016	0.9919	0.956 ± 0.011	0.261 ± 0.064
Color parameters						
L*	MSC-DV1	10	0.385	0.9915	0.967 ± 0.011	1.757 ± 0.594
a*	MSC-SM	10	0.591	0.9580	0.934 ± 0.024	1.064 ± 0.399
b*	DV1	10	0.129	0.9999	1.000 ± 0.001	ns
C*	SNV-SM	10	0.528	0.9664	0.941 ± 0.022	1.186 ± 0.443
Chemical parameters						
Protein (%)	ALS-SM	10	0.116	0.9867	0.956 ± 0.014	0.935 ± 0.293
FA (%)						
C14:0	DV2	10	0.125	0.9927	0.938 ± 0.010	0.284 ± 0.049
C16:0	DV2	10	0.223	0.9918	0.956 ± 0.011	1.134 ± 0.276
C16:1n-7	MSC-SM	30	0.050	0.9995	1.000 ± 0.003	ns
C17:1n-7	DV2	10	0.012	0.9908	0.968 ± 0.012	0.020 ± 0.007
C18:0	MSC-SM	30	0.176	0.9915	0.971 ± 0.0111	0.334 ± 0.134
C18:0	NORM-DV1	10	0.189	0.9998	0.999 ± 0.002	ns
9t-C18:1	MSC-DV1	10	0.028	0.9921	0.950 ± 0.010	0.077 ± 0.017
C18:1n-9	DV2	10	0.394	0.9913	0.966 ± 0.011	1.315 ± 0.440
C18:3n-3	MSC-DV1	10	0.014	0.9916	0.956 ± 0.011	0.029 ± 0.008
∑MUFA	MSC-DV1	10	0.406	0.9910	0.972 ± 0.012	1.226 ± 0.517
IA	DV2	10	0.016	0.9926	0.939 ± 0.010	0.050 ± 0.008
h/H	DV2	10	0.026	0.9924	0.943 ± 0.010	0.097 ± 0.017

SM—smoothing; DV1—first derivative; DV2—second derivative; NORM—normalization; SNV—standard normal variate correction; MSC—multiplicative scatter correction; ALS—asymmetric least squares; RMSE—root-mean-square error; R^2^—coefficient of determination; s—standard deviation; *—train slope *p*-value < 0.001; ns—not significant; (L*)—luminosity index; (a*)—red index; (b*)—yellow index; (C*)—chroma; FA—fatty acid; MUFA—monounsaturated fatty acid; IA—index of atherogenecity; h/H—hypocholesterolemic/hypercholesterolemic index.

**Table 5 foods-13-01581-t005:** The predictive performance of the linear support vector regression model on the test data.

Parameters	Data Treatment	Best C	RMSE	R^2^	Slope ± s *
Physical parameters					
pH (24 h)	NORM-DV1	10	0.221	0.9986	0.9805 ± 0.057
Color parameters					
L*	MSC-DV1	10	4.243	0.9939	1.007 ± 0.034
a*	MSC-SM	10	3.200	0.9697	1.031 ± 0.007
b*	DV1	10	1.675	0.9822	0.960 ± 0.014
C*	SNV-SM	10	2.910	0.9837	1.037 ± 0.076
Chemical parameters					
Protein (%)	ALS-SM	10	1.112	0.9975	1.029 ± 0.022
FA (%)					
C14:0	DV2	10	1.232	0.9389	0.845 ± 0.0510
C16:0	DV2	10	2.708	0.9896	1.016 ± 0.026
C16:1n-7	MSC-SM	30	0.742	0.9059	0.911 ± 0.033
C17:1n-7	DV2	10	0.112	0.9669	0.970 ± 0.027
C18:0	MSC-SM	30	2.518	0.9467	0.913 ± 0.023
C18:0	NORM-DV1	10	1.714	0.9750	0.920 ± 0.019
9t-C18:1	MSC-DV1	10	0.369	0.9406	0.832 ± 0.114
C18:1n-9	DV2	10	3.953	0.9906	1.048 ± 0.047
C18:3n-3	MSC-DV1	10	0.198	0.9205	0.968 ± 0.024
∑MUFA	MSC-DV1	10	5.271	0.9878	1.056 ± 0.018
IA	DV2	10	0.168	0.9610	0.915 ± 0.018
h/H	DV2	10	0.346	0.9560	1.059 ± 0.023

SM—smoothing; DV1—first derivative; DV2—second derivative; NORM—normalization; SNV—standard normal variate correction; MSC—multiplicative scatter correction; ALS—asymmetric least squares; RMSE—root-mean-square error; R^2^—coefficient of determination; s—standard deviation; *—test slope *p*-value < 0.001; ns—not significant; (L*)—luminosity index; (a*)—red index; (b*)—yellow index; (C*)—chroma; FA—fatty acid; MUFA—monounsaturated fatty acid; IA—index of atherogenecity; h/H—hypocholesterolemic/hypercholesterolemic index.

## Data Availability

The original contributions presented in the study are included in the article/Appendix A, further inquiries can be directed to the corresponding author.

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
