# Peer review of "Contribution to Characterizing the Meat Quality of Protected Designation of Origin Serrana and Preta de Montesinho Kids Using the Near-Infrared Reflectance Methodology"

_foods, 2024, doi:10.3390/foods13101581_

Round 1

Reviewer 1 Report

Comments and Suggestions for Authors

The article is well written and easy to read.
The Abstract deserves a slight rewrite to make it easier to read, with fewer numbers and abbreviations, to explain more about the interest of this study and the strategy employed.
The Introduction section needs to be shortened to explain the prediction methods more fully, what criteria are generally used to predict meat quality, on what number of animals and with what validation strategy.
The Materials and Methods section is very detailed and provides all the information needed to reproduce the experiments and calculations.
The Results section is very clear and well presented. It is also well referenced, which is pleasing and supports the results obtained here.
Overall, the article need a few simplifications to make it easier to read, as explained in the summary.

Author Response

Dear review,

All modifications were made following the reviewer`s suggestions and comments, and

responses to their comments are also attached. Thanks to their recommendations,

significant modifications were made throughout the manuscript.

Thank you for your attention.

Answers to Reviewer 1

Comments and Suggestions for Authors

The article is well written and easy to read.

Response: We thank the reviewer for his attention to the review of the article.

The Abstract deserves a slight rewrite to make it easier to read, with fewer numbers and abbreviations, to explain more about the interest of this study and the strategy employed.

Response: Several changes have been made in the abstract according to the reviewer`s comments.

The Introduction section needs to be shortened to explain the prediction methods more fully, what criteria are generally used to predict meat quality, on what number of animals and with what validation strategy.

Response: We understand that the introduction could be lengthy by introducing information on goat breeds. However, we believe that much of it is necessary for readers to understand the context of meat quality labels. Nevertheless, we have deleted some sentences regarding kid meat quality and added in the text the following sentences regarding NIR according to the reviewer`s suggestion.

Added

“The spectrometric technologies are gaining importance as tools of meat quality control to effectively predict carcass and meat quality traits in the industry. Among them, (…)”

“This fast, reliable and non-destructive technology can measures the absorption of electromagnetic radiation in the near-infrared spectrum (750–2500 nm), however it requires a chemometric methods model able to relate the spectral absorption with analytical data obtained from a reference method of chemical properties [12,13].”

“Additionally, NIR also requires a multivariate statistical technique, such as principal component analysis (PCA), projections algorithm (SPA), and k-nearest neighbors’ algorithm (KNN), for qualitative exploratory analysis. Also, it`s applied data modeling’s such linear discriminant analysis (LDA), partial least square discriminant analysis (PLS-DA) and support vector ma-chine (SVM) to the sample’s identification, characterization and discrimination. The SVM, partial least square regression (PLSR) and artificial neural network (ANN) are also used for quantitative purposes [11, 17]. However, to develop reliable calibration models, a validation process is required for successful results. The validation method commonly used is the internal or cross-validation, or external validation, the latter of which can provide more reliable and relevant estimates of the model's future predictive capacity but is not advisable in biological samples or in small numbers [12]. So, it is essential choosing the samples for inclusion in the calibration set, determining the property to be predicted by using an appropriate method to measure such samples, obtaining the analytical spectral signal, constructing the model, validating it and, finally using it to predict the samples. Generally, the coefficient of determination (R2), root mean square error (RMSE) and/or ratio of performance deviation (RPD) are criteria generally used to predict meat quality [12]. For these reasons, the study of these breeds and their characterization is fundamental to deepening knowledge and enhancing the quality of “Cabrito Transmontano”, as well as reinforcing the use of cutting-edge methodologies in an efficient and quick way to estimate the quality attributes of this important meat with a protected designation of origin which is the aim of this study.”

The Materials and Methods section is very detailed and provides all the information needed to reproduce the experiments and calculations.

The Results section is very clear and well presented. It is also well referenced, which is pleasing and supports the results obtained here.

Overall, the article need a few simplifications to make it easier to read, as explained in the summary.

Response: We have revised the manuscript to simplify the reading and we would like to thank the reviewer for the attention given to reviewing the article and for the comments, which we have used to improve some areas of the manuscript as suggested.

Reviewer 2 Report

Comments and Suggestions for Authors

This study evaluated the quality traits and  NIR spectra of suckling kid meat from "the Serrana" and "Preta de Montesinho". It was revealed that the breeds had the dominant influnence on the properties of meat. This work has some reality significance but lacks of logicality. I would suggest authors check the experiments, polish the whole manuscript and resubmitted.

1.The age of femals kids from Preta de Montesinho was 67 days, which were different from other samples. Although the age seems to have no effect on the weight and yield, it might influence the properties of meat.

2.Authors purchased a total of 16 kids per breed. The characterizations were conducted on every sample individually or the mixture of 16 samples? Please add the details in the experimental part.

3 The amounts of samples were not sufficient for prediction. 

4 In the experimental procedures of NIR, the mixture mince of 32 samples were used. How did authors establish NIR predictive models to discriminate between the two breeds based on the NIR data from the mixture of two breeds?

5 The properties of meat were usually characterized in replicates. There seems no replicates in this work.

Comments on the Quality of English Language

There are some spelling and grammar errors. Authors should check and polish the manuscript carefully.

Author Response

Dear review,

All modifications were made following the reviewer`s suggestions and comments, and

responses to their comments are also attached. Thanks to their recommendations,

significant modifications were made throughout the manuscript.

Thank you for your attention.

Answers to Reviewer 2

Comments and Suggestions for Authors

This study evaluated the quality traits and NIR spectra of suckling kid meat from "the Serrana" and "Preta de Montesinho". It was revealed that the breeds had the dominant influence on the properties of meat. This work has some reality significance but lacks of logicality. I would suggest authors check the experiments, polish the whole manuscript and resubmitted.

Response: We thank the reviewer for the attention given to review the article.

1.The age of females kids from Preta de Montesinho was 67 days, which were different from other samples. Although the age seems to have no effect on the weight and yield, it might influence the properties of meat.

Response: Our experiment was designed to sample animals inside the age and weight ranges stipulated in quality label, i.e. slaughtered between 7 and 12 kg in weight and at 60-80 days of age. The result of carcass purchasing give a lower mean values (5-6 days lower) for Preta de Montesinho females compared to the other group; however, the differences between each individual groups were not significant. The low difference in days and specially the lack of significance together with the fact that all animals were into the ranges of the regulation allows us to assume that the populations are sufficiently homogeneous in terms of age.

2.Authors purchased a total of 16 kids per breed. The characterizations were conducted on every sample individually or the mixture of 16 samples? Please add the details in the experimental part.

Response: Yes, the characterizations were conducted on every sample individually. This information has been addressed to the revision manuscript.

3 The amounts of samples were not sufficient for prediction.

Response: We concur with the reviewer that the sample size is inadequate and that additional samples would be crucial to reinforce the model's robustness. Nevertheless, the primary aim of this study is to illustrate that the outcomes obtained herein underscore the methodology's efficacy in predicting test data (data not utilized in the calibration model's derivation). It is noteworthy that as long as the test data falls within the range defined by the training data, one can infer that predictive capability will be sustained.

4 In the experimental procedures of NIR, the mixture mince of 32 samples were used. How did authors establish NIR predictive models to discriminate between the two breeds based on the NIR data from the mixture of two breeds?

Response: From the analysis of the clarification request, it became apparent that the information contained within the article was limited, and misdirected.

So, hence, a revised text was submitted to clarify this issue the text in section “2.5 Chemometrics and Data Analysis”:

“The sample spectra were divided into calibration set (training group - 20 samples (75% of samples), each subjected to three repetitions of NIR spectral analysis, conducted from three distinct positions, in order to encompass the variability inherent in meat sample) and prediction set (test group - 10 samples (25% of samples) with 3 spectrum repetitions of NIR analysis). The data partitioning was conducted using the Kennard-Stone algorithm, with Euclidean metric, leveraging the physicochemical attributes of the meat samples [35,36] to delineate these two distinct data groups, common across all data treatments. These two groups were used in two data treatments (discrimination and quantification) by applying the linear support vector machine (SVM) techniques that does not depend on the distributions of the underlying dependent and independent variables.” 

5 The properties of meat were usually characterized in replicates. There seems no replicates in this work.

Response: All the meat quality analyses contained in the study were carried out in triplicate (pH, water activity, shear force, instrumental color, humidity, ash, protein, lipid profile). This information is contained and reinforced in the manuscript:

“All analysis were performed in triplicate for each method and each animal.”

“Three spectra were measured for each LT sample, and the average of these spectra was used in further chemometric analysis.”

“The three spectra replicates were used in the analysis since it allowed us to include the variability associated with heterogeneity of the samples contributing to the adjustment of more robust models.”

Also, in NIR modeling, it is acceptable for the physicochemical attributes of meat samples to be represented by average values, as they reflect the outcomes derived from reference method analyses of individual samples. However, when considering NIR spectra, repetitions are crucial to accommodate the inherent variability within meat samples. This is particularly important given the technique's objective of facilitating in situ meat analysis on processing lines, where variability must be accounted for in the modeling process.

A new phrase was introduced in section “2.3 Near-Infrared Spectroscopy” to clarify this issue:

“These repetitions were aimed at accommodating the intrinsic variability within meat samples. This is especially critical considering the technique's aim of enabling in situ meat analysis on processing lines, where variability must be incorporated into the modeling process.”

Comments on the Quality of English Language

There are some spelling and grammar errors. Authors should check and polish the manuscript carefully.

Response: A careful revision of the English was carried out to improve it.

Reviewer 3 Report

Comments and Suggestions for Authors

This manuscript evaluated the meat quality characteristics of protected designation of origin (PDO) Serrana and Preta de Montesinho kids using the near infrared reflectance (NIR) methodology. Because of minor problems this manuscript requires revision.

1. In Abstract section, what is the full name of FA? Please provide it on the first appearance.

2. In “2.2 Physicochemical analysis and chemical composition of meat and fat” section, I suggest the authors list the subheadings, for example pH value, color, water holding capacity, …….

3. Line170, “K-type probe; Therma 3, Industrial Thermometer.” Please the authors provide the model and company name of K-type probe, as well as city and country where the company is located.

Author Response

Dear review,

All modifications were made following the reviewer`s suggestions and comments, and

responses to their comments are also attached. Thanks to their recommendations,

significant modifications were made throughout the manuscript.

Thank you for your attention. 

Answers to Reviewer 3

Comments and Suggestions for Authors

This manuscript evaluated the meat quality characteristics of protected designation of origin (PDO) Serrana and Preta de Montesinho kids using the near infrared reflectance (NIR) methodology. Because of minor problems this manuscript requires revision.

Response: We are thankful the reviewer for the attention given to reviewing the article, which we took advantage of to improve the manuscript.

  1. In Abstract section, what is the full name of FA? Please provide it on the first appearance.

Response: Changes have been made in revision article

  1. In “2.2 Physicochemical analysis and chemical composition of meat and fat” section, I suggest the authors list the subheadings, for example pH value, color, water holding capacity, …….

Response: We are thankful for the suggestions; We have made four subsections in this topic to be clearer and more organized.

  1. Line170, “K-type probe; Therma 3, Industrial Thermometer.” Please the authors provide the model and company name of K-type probe, as well as city and country where the company is located.

Response: Changes have been made in the revised version of the manuscript to provide the model, brand, city and country.

Reviewer 4 Report

Comments and Suggestions for Authors

Dear, in the following manuscript, the authors tackle to topic of Contribution to characterizing the meat quality of protected designation of origin (PDO) Serrana and Preta de Montesinho kids using the NIR methodology.

Food authenticity and food traceability are of great concern to
consumers, food processors, retailers, and regulatory bodies. Therefore, the topic is in line with the food strategy as part of its food quality policy followed by reducing food fraud and the risk of fraudulent labeling. Moreover, the demonstration of authenticity of high-value meat products is of great interest and importance within the food sector, in addition to increasing demand for fast and low-cost analytical techniques for quality control in an industrial setting.

Therefore, I recommend this article for major revision. The manuscript is well-organized and written. My suggestion is addressed to quality of presentation. Presentation quality can be improved by employing PCA to visualize differences between analyzed samples better. Also, the conclusion must be improved to emphasize the practical and technological relevance and importance of research.

Author Response

Dear review,

All modifications were made following the reviewer`s suggestions and comments, and

responses to their comments are also attached. Thanks to their recommendations,

significant modifications were made throughout the manuscript.

Thank you for your attention.

Answers to Reviewer 4

Comments and Suggestions for Authors

Dear, in the following manuscript, the authors tackle to topic of Contribution to characterizing the meat quality of protected designation of origin (PDO) Serrana and Preta de Montesinho kids using the NIR methodology.

Food authenticity and food traceability are of great concern to consumers, food processors, retailers, and regulatory bodies. Therefore, the topic is in line with the food strategy as part of its food quality policy followed by reducing food fraud and the risk of fraudulent labeling. Moreover, the demonstration of authenticity of high-value meat products is of great interest and importance within the food sector, in addition to increasing demand for fast and low-cost analytical techniques for quality control in an industrial setting.

Therefore, I recommend this article for major revision. The manuscript is well-organized and written. My suggestion is addressed to quality of presentation. Presentation quality can be improved by employing PCA to visualize differences between analyzed samples better. Also, the conclusion must be improved to emphasize the practical and technological relevance and importance of research.

Response:  Dear reviewer, thank you for your appreciation and consideration of the work under study.  In fact, multivariate classification is one of the basic methodologies in chemometrics and consists in finding mathematical relationships between a set of descriptive qualitative variables. It is clear that the development of reliable methods for to guarantee authenticity is becoming very important in order to authenticate the origin and differentiate it with regard to chemical and physical parameters wherever possible.

Principal Component Analysis (PCA) applied to NIR spectra may not yield satisfactory results for discrimination purposes due to several factors. Firstly, NIR spectra often contain overlapping peaks and complex spectral patterns, making it challenging to separate classes effectively based solely on spectral variation. Additionally, the underlying physicochemical differences between classes may not be well-captured by the principal components extracted from the spectra, leading to poor discrimination performance. Furthermore, the non-linear nature of the relationship between the spectral data and the discriminating factors can hinder the ability of PCA to effectively separate classes in multidimensional space. Therefore, while PCA can be useful for dimensionality reduction and visualization of spectral data, alternative methods such as supervised classification techniques may be more suitable for discrimination tasks in NIR analysis.

To elucidate this point, the following are the plots of the first two principal components of all spectral transformations utilized in this study. The plots reveal consistent overlap between the two goat breeds, necessitating the application of a supervised discrimination technique for enhanced separation efficiency.

However, should the reviewer deem this information relevant to the article, we can proceed with its inclusion.

Regarding the conclusion we have revised it and made changes according the reviewer’s comments.

Round 2

Reviewer 2 Report

Comments and Suggestions for Authors

The manuscipt has been improved after careful revision. However, there are still some issues to be addressed before acceptance.

1Lin 90: Change "reared an extensive" to "reared in an extensive"

2 Line 111: Change "This way" to "In this way".

3 Line 129: Change "very little" to "rarely".

4 The background of this study was not dipicted clearly. The origin and breeds were introduced in the second and third paragraph , which was followed by the introduction of  spectrometric technologies. The writing is tangled and lacks of coherence. The importance and contribution of the  spectrometric technologies were neglected.

5 Line 123-158: This paragraph can be divided into two paragraphs. It lacks of the work intended to do and the significance of this study.

6 Line 279: What is "LTL"? No explanation for this abbreviation. "A  total  of  32  samples  of  LTL  were  minced  using  a  power  mill  Buchi  Mixer  B-400 and placed in petri dishes with a diameter of approximately 9 cm."  32 samples were minced and characterized by FT-NIR individually or totally? Please explain and add more details. 

7 The format of the references was not consistent. Please check and revise it.

Comments on the Quality of English Language

There are still some grammar errors. The quality of English language can be improved.

Author Response

Dear review,

All modifications were made following the reviewer`s suggestions and comments, and

responses to their comments are also attached. Thanks to their recommendations,

significant modifications were made throughout the manuscript.

Thank you for your attention.

Answers to Reviewer 2

Comments and Suggestions for Authors

The manuscipt has been improved after careful revision. However, there are still some issues to be addressed before acceptance.

1Lin 90: Change "reared an extensive" to "reared in an extensive"

Response: Suggestion accepted; changes have been made in the revised version of the

manuscript.

2 Line 111: Change "This way" to "In this way".

Response: Suggestion accepted; changes have been made in the revised version of the

manuscript.

3 Line 129: Change "very little" to "rarely".

Response: Suggestion accepted; changes have been made in the revised version of the

manuscript.

4 The background of this study was not dipicted clearly. The origin and breeds were introduced in the second and third paragraph, which was followed by the introduction of spectrometric technologies. The writing is tangled and lacks of coherence. The importance and contribution of the spectrometric technologies were neglected.

Response: In order to clarify the importance and contribution of the study, particularly spectrometric technologies, changes have been made to the review manuscript.

5 Line 123-158: This paragraph can be divided into two paragraphs. It lacks of the work intended to do and the significance of this study.

Response: Suggestion accepted; changes have been made and information have been added in the revised version of the manuscript.

6 Line 279: What is "LTL"? No explanation for this abbreviation. "A total  of  32  samples  of  LTL  were  minced  using  a  power  mill  Buchi  Mixer  B-400 and placed in petri dishes with a diameter of approximately 9 cm."  32 samples were minced and characterized by FT-NIR individually or totally? Please explain and add more details.

Response: We apologized for mistake, the sigla “LTL” contained an error. The correct form is “LT” which corresponding to Longissimus thoracis (LT) muscle. The 32 samples were minced and characterized individually. To be clearer the changes have been made in the revised version of the manuscript.

7 The format of the references was not consistent. Please check and revise it.

Response: Suggestion accepted; changes have been made in the revised version of the

manuscript.

Comments on the Quality of English Language

There are still some grammar errors. The quality of English language can be improved.

Response:  The manuscript was revised in order 

Reviewer 4 Report

Comments and Suggestions for Authors

Dear, after carefully considering of revised manuscript I can conclude that author provided satisfactory answers on my suggestion.
Extensive changes have been made and improved manuscript accordingly. As before regarding to quality of investigation and presentation of results I can recommend this article for publication.

Author Response

Dear review,

All modifications were made following the reviewer`s suggestions and comments, and

responses to their comments are also attached. Thanks to their recommendations,

significant modifications were made throughout the manuscript.

Thank you for your attention.

Answers to Reviewer 4

Dear, after carefully considering of revised manuscript I can conclude that author provided satisfactory answers on my suggestion.

Extensive changes have been made and improved manuscript accordingly. As before regarding to quality of investigation and presentation of results I can recommend this article for publication.

Response: Dear review, with the help of your suggestions and comments, we have been able to improve the manuscript in terms of the quality of the research and the presentation of the results for future readers. Thank you for your work.

Best regards,

the authors
